# Hot Deformation Behavior and Microstructure Evolution of Near-α Titanium Alloy TA32 in Dual-Phase Zone

**DOI:** 10.3390/ma18071476

**Published:** 2025-03-26

**Authors:** Jiajun Jiang, Yi Meng, Yingxu Cheng, Ruiqi Wang, Xingang Liu

**Affiliations:** 1College of Materials and Engineering, Chongqing University, Chongqing 400044, China; mengyi@cqu.edu.cn; 2China National Erzhong Group Deyang Wanhang Die Forging Co., Ltd., Deyang 618000, China; 18808106446@163.com; 3School of Mechanical Engineering, Yanshan University, Qinhuangdao 066004, China; scj@stumail.ysu.edu.cn (Y.C.); lxg@ysu.edu.cn (X.L.)

**Keywords:** TA32 titanium alloy, constitutive equation, hot deformation, microstructure evolution, dynamic recrystallization

## Abstract

The hot deformation behavior of the near-α titanium alloy TA32 (Ti-5.5Al-3.5Sn-3Zr-1Mo-0.5Nb-0.7Ta-0.3Si) was studied by isothermal compression tests. The deformation temperatures ranged from 700 to 950 °C, with strain rates ranging from 0.001 to 1.0 s⁻¹. The stress–strain curves corresponding to different deformation parameters were studied to evaluate the mechanical behavior. A prediction model of peak stress of TA32 titanium alloy in the dual-phase zone was established, utilizing friction-temperature-corrected flow stress. Electron backscatter diffraction (EBSD) and scanning electron microscopy (SEM) were used to examine the influence of deformation parameters on microstructure evolution. The findings reveal that at 950 °C and 0.01 s⁻¹, the recrystallized volume fraction reaches 34.6%, with an average recrystallized grain size measuring 3.03 μm, which is significantly enhanced compared with those at lower deformation temperatures. By examining the softening behavior across different deformation parameters, it was concluded that dynamic recrystallization (DRX) becomes the primary mechanism. The conclusions of this study can provide some reference and guidance for the microstructure evolution of TA32 alloy during the hot deformation process so as to accelerate the design and optimization of deformation process parameters and the development and popularization of a new high-temperature titanium alloy TA32.

## 1. Introduction

Titanium alloys are widely utilized in chemical, energy, biomedical, and aerospace fields due to their notable characteristics, including superior heat resistance, outstanding ductility, and high specific strength [1,2,3,4,5]. Near-α titanium alloys have numerous outstanding properties, including excellent high-temperature creep resistance and good welding abilities [6,7,8,9], which can ensure the stability of work in the temperature range of 400~600 °C [10].

Thermal processing is the primary method for optimizing the performance and adjusting the microstructure to match the service requirements of the workpiece [11,12,13,14]. The forming temperature range of titanium alloys is narrow, and their deformation resistance and microstructure are highly responsive to variations in thermal deformation conditions, including strain rate, deformation temperature, and strain. Thermal processing typically involves intricate hardening and softening behaviors, which complicates the control of the hot working process [15]. Therefore, the thermal deformation and microstructure evolution of relations are of great significance in guiding the machining process, controlling the properties, and improving the microstructure of titanium alloys [16].

The deformation behavior during thermal processing is represented by the connection between deformation parameters and flow stress. This connection is described by the material’s constitutive equation [17]. The constitutive equation is a mathematical expression to describe the dynamic response between the thermal parameters of the material in the process of high-temperature deformation [18]. *Q*, *A*, and *n* can be obtained by constructing the Arrhenius equation of titanium alloy. These parameters are essentially the macroscopic embodiment of the microstructure evolution mechanism. The thermal activation energy *Q* is the minimum energy required to overcome the energy barrier during thermal deformation and can reflect the difficulty of dislocation motion, DRX, or phase transition mechanisms. The higher the proportion of recrystallized grains, the *Q* may decrease because after DRX startup, new grains form and consume dislocation, resulting in a decrease in *Q* value. The α phase content in the equiaxed structure is higher, and the material constant *A* value is usually larger because of the uniform deformation of the equiaxed structure, and the α phase is harder than the β phase. Dynamic recovery or DRX results in dislocation annihilation, and the stress index *n* value may decrease due to reduced strain rate sensitivity. The flow behavior observed in the process of thermal deformation is a direct physical reflection of the microstructure evolution. The parameters of these constitutive equations can comprehensively reflect the dislocation movement and recrystallization in the microstructure and are closely related to the microstructure state and dynamic evolution. Therefore, Arrhenius-type equations are widely used in the analysis of metallic materials’ deformation behavior due to their high efficiency and accuracy in predicting high-temperature rheological stresses [19,20]. Wang et al. [21] applied the Arrhenius-type model to develop an intrinsic model of Ti-0.3Mo-0.8Ni-2Al-1.5Zr alloy in the dual-phase zone. The results confirmed that the experimental results showed a strong correlation with the predicted peak stress values.

Although the Arrhenius equations in hyperbolic sinusoidal form have been commonly utilized to estimate flow stress in titanium alloys, the impact of strains is not taken into account, and the prediction results have some deviation. Therefore, Arrhenius-type models with strain compensation are widely used [22]. Lu et al. [23] evaluated the deformation features of Ti-35421 alloy in the dual-phase zone and established, respectively, the two intrinsic models with or without considering strain compensation. It was found by comparison that the intrinsic equations would predict the flow stresses more precisely when considering strains. Chai et al. [24] established Arrhenius-type equations with strain compensation correction and realized the fast and accurate prediction of flow stresses under varying deformation conditions of TC11 alloy.

In this paper, thermal compression experiments of TA32 alloy have been carried out with different hot deformation parameters, and friction-temperature corrections were made to the flow stresses obtained. Using the corrected true flow curve, a model was constructed to forecast the peak stress in the dual-phase zone. The microstructure after deformation was characterized using the EBSD test, and the dynamic recrystallization behavior was analyzed to determine the discontinuous dynamic recrystallization (DDRX) and continuous dynamic recrystallization (CDRX) behaviors. The results provide certain reference and theoretical values for the hot deformation process of TA32 alloy. It also helps to accelerate the popularization and application of TA32 titanium alloy.

## 2. Materials and Methods

### 2.1. Materials Preparation

The material used in this study was a novel near-α high-temperature titanium alloy provided by China National Erzhong Group Deyang Wanhang Die Forging Co., Ltd. (Deyang, China). This alloy exhibits significant potential as a candidate material for manufacturing compressor blades and other critical structural components in aero-engines, owing to its excellent high-temperature mechanical properties and thermal stability. The chemical composition provided by the supplier of the alloy is shown in Table 1, and the phase transition temperature of the alloy is approximately 995 ± 5 °C according to the continuous heating metallographic method. The thermal compression experiment on TA32 alloy was carried out utilizing the Thermmecamastor-z (Kawasaki, Japan) thermal simulation tester. The compression direction of the samples corresponds to the forging direction of the as-received billet.

### 2.2. Thermomechanical Processing Experiments

Figure 1 describes the microstructure observation region and the thermal compression process route. The raw material was cut into cylindrical specimens of standard size Φ8 × 12 mm utilizing wire electrical discharge machining (WEDM). To lessen friction between the specimen and the indenter, tungsten carbide shims were positioned between the top and lower indenters. During the testing process, the equipment needs to be vacuumed to prevent the specimen surface from oxidizing. The specimen is heated by an electromagnetic induction coil, and a computer system is used to collect information on load, displacement, temperature, stress, and strain. The temperature was increased at a 10 °C/s rate and maintained for five minutes to ensure thermal uniformity. The specimen was cooled with helium immediately after deformation. Six deformation temperatures (700 °C, 750 °C, 800 °C, 850 °C, 900 °C, 950 °C) and four deformation rates (0.001 s^−1^, 0.01 s^−1^, 0.1 s^−1^, 1.0 s^−1^) were utilized in the experiments, with thermally compressed specimens deforming by 60%.

### 2.3. Microstructure Characterization

To observe the crystallographic orientation information of the specimens after thermal deformation, the specimen was sliced parallel to the compression direction, and the center region was selected for observation. The cut specimens were ground on an automatic grinding and polishing machine using 600 #, 1200 #, and 2500 # waterproof water sandpapers until there were no obvious scratches. They were polished using an Al_2_O_3_ suspension and a diamond polishing solution, respectively. The Zeiss Sigma 300 (Jena, Germany) scanning electron microscope were utilized to collect EBSD data; the electron beam voltage was 20 kV, and the scanning step was 0.5 μm. The EBSD data were analyzed with the AZtec Crystal (AZtecCrystal 2.1) analysis software to determine the impact of deformation parameters on the microstructure.

## 3. Results and Discussion

### 3.1. Initial Microstructure

The initial microstructure of TA32 alloy is illustrated in Figure 2. From Figure 2, the microstructure of TA32 alloy contains primary equiaxed α_p_ and well-arranged lamellar secondary α_s_ in the β-transformed matrix. Statistically, the content of equiaxed αp in the initial microstructure accounts for about 43.4%, and the average grain size reaches 19.6 μm, indicating a typical bimodal structure. It can be seen that the lamellar αs phase within the transformed β matrix exhibited a characteristic basket-weave morphology. This microstructure configuration, combining equiaxed α_p_ grains with lamellar α_s_ structure, is particularly advantageous for achieving an optimal balance between strength and ductility in titanium alloys.

### 3.2. Constitutive Equation

#### 3.2.1. Stress–Strain Curve

The stress–strain curves of TA32 alloy at different temperatures with the same deformation rate are presented in Figure 3. At the beginning of deformation, the effect of work hardening causes the flow stress to rapidly climb to its highest value as strain increases [25]. With further strain, the flow stress begins to decline gradually after reaching its maximum, signaling the onset of alloy softening. Finally, the work hardening and dynamic softening are balanced dynamically. This indicates that the DRX appears in the hot deformation process, which is most noticeable at lower strain rates and temperatures [26]. Additionally, the peak stress falls as the deformation rates reduce. The long deformation time is the cause of this, where dislocation accumulation is attenuated, resulting in a reduction in flow stress [27]. At a 1.0 s^−1^ strain rate, the peak stress reduced from 658.13 MPa to 146.69 MPa as the deformation temperature increased from 700 to 950 °C. This occurs because the interatomic bonding in the alloy is weakened, and the deformation proceeds more easily. Secondly, the deformation resistance between the alloy grains becomes smaller as temperature increases, which provides power for the movement of dislocations, and slip is more likely to happen.

#### 3.2.2. Friction Correction

During the hot compression process, titanium alloy specimens exhibit bulging after isothermal compression, a phenomenon primarily caused by friction between the compression heads and the specimen. To address this phenomenon, the bulging coefficient B is used to evaluate the influence of friction on flow stress, and the decision to correct the flow stress curve for friction is based on the magnitude of B. The specific expression for the bulging coefficient B is as follows:(1)$$B=\frac{h_{1}R_{M}^{2}}{h_{0}R_{0}^{2}}$$
where *h*_1_ is the height of the specimen after deformation; *R*_M_ is the maximum bulging radius; *h*_0_ is the initial height of the specimen; and *R*_0_ is the initial diameter of the specimen. When 1 < *B* < 1.1, the influence of friction on flow stress is minimal, and no correction to the experimental data is necessary. When *B* > 1.1, it indicates that friction has a significant impact on flow stress, leading to a certain error between the measured data and the true stress and necessitating friction correction. Table 2 shows the bulging coefficient *B* for TA32 alloy under different deformation conditions. It can be observed that the *B* values for the tests are almost all greater than 1.1, making it essential to perform friction correction on the flow stress.

Ebrahimi et al. [28] proposed that the experimentally obtained true stress–strain curve data can be corrected using the friction correction factor *μ*. The relationship between the friction correction factor *μ* and the dimensions of the specimen during hot deformation is expressed by the following equation:(2)$$\sigma =\frac{C^{2}\sigma_{0}}{2\left(e^{C}-C-1\right)}$$(3)$$\mu =\frac{\left(R/H\right)b}{4/\sqrt{3}-\left(2b/3\sqrt{3}\right)}$$(4)$$b=4\frac{∆R}{R}\cdot \frac{H}{∆H}$$
where $R=R_{0}\sqrt{H_{0}/H}$ and *H* are the average radius and height of the specimen after hot compression, respectively; *R*_0_ and *H*_0_ are the radius and height of the specimen before hot compression, respectively; *b* is the cylindrical parameter; Δ*R* is the difference between the maximum radius *R_M_* and the top radius *R_T_* of the specimen after hot compression (Δ*R* = *R_M_*
− *R_T_*); Δ*H* is the height difference of the specimen before and after hot compression (Δ*H* = *H* − *H*_0_); σ is the flow stress value after friction correction; σ_0_ is the experimentally measured flow stress value; and $C=\left(2\mu R_{0}\right)/H_{0}$ represents a factor related to geometric structure. Using the above equation, the flow stress after friction correction can be calculated. Figure 4 shows a comparison between the experimentally measured flow stress curves and the friction-corrected flow stress curves of TA32 titanium alloy under different deformation conditions. It can be observed that the flow stress values decrease after friction correction, and the difference between the measured values and the true values gradually increases with increasing strain. This is because, under the same deformation conditions, an increase in strain leads to an increase in the contact area between the specimen and the equipment, enhancing the friction effect and increasing the deformation resistance. As a result, the experimentally measured flow stress values increase more significantly.

#### 3.2.3. Temperature Correction

TA32 alloy easily produces an adiabatic temperature rise effect during deformation because of its poor heat dissipation property. The adiabatic temperature rise phenomenon during compression is primarily related to the strain rate. At high strain rates, the specimen can be approximated as an adiabatic body, resulting in a significant temperature rise. However, when the strain rate is less than 0.01 s^−1^, the heat generated by deformation is relatively small, and there is sufficient time for it to dissipate through conduction, making correction unnecessary. The temperature variation ∆T can be calculated using Equation (5):(5)$$∆T=\frac{0.95\eta \int_{0}^{\varepsilon }\sigma d\varepsilon }{\rho C_{p}}$$
where $∆T,\;\rho ,C_{p},\;\int_{0}^{\varepsilon }\sigma d\varepsilon$ are temperature variation, material density, specific heat capacity, and mechanical work, respectively. *η* is the adiabatic correction coefficient, which the strain rate determines (when $\dot{\varepsilon }$ ≤ 0.001 s^−1^, *η* = 0; when $\dot{\varepsilon }$ ≥ 10 s^−1^, *η* = 1; when 0.01 s^−1^ ≤ $\dot{\varepsilon }$ ≤ 1.0 s^−1^, *η* = 0.25lg($\dot{\varepsilon }$) + 0.75). For TA32 alloy, ρ =4.51 g/cm^3^, =0.87 J/(g·K). Figure 5a describes the value of the deformation temperature change at different strain rates. It is obvious that the temperature change value reduces as the deformation temperature rises. At the same deformation temperature, the value of temperature change rises as the strain rate increases. This is due to the temperature rise generated by hot deformation having enough time to heat exchange with the external medium at lower deformation rates. The short deformation time hinders effective heat exchange and dissipation, leading to an increase in the temperature rise.

The flow curves before and following the application of the friction-temperature correction under different thermal deformation parameters are shown in Figure 5b–d. It is evident that as the deformation temperature increases, the difference in flow stress progressively decreases. The adiabatic temperature rise effect is obvious at high deformation rates, and the increase in flow stress is larger after temperature correction. As deformation temperatures decrease, the value of the corrected change becomes more significant. This is because the temperature increase can provide power for deformation, thus reducing the heat generated during thermal deformation. Faster strain rates lead to greater adiabatic temperature rise effects, where the heat generated does not have time to diffuse, ultimately leading to greater stress differentials. The modified flow curve can eliminate the effect of deformation temperature and friction to a certain extent and is closer to the real rheological curve.

#### 3.2.4. Peak Stress Prediction Model

To accurately describe the flow stress of TA32 titanium alloy, a peak stress prediction model based on friction-temperature correction was developed. Tegart and Sellars [29] introduced the hyperbolic sine equation to represent the connection between strain rate, temperature, and strain, incorporating both temperature and deformation activation energy.(6)$$\dot{\varepsilon }=A_{1}\sigma^{n_{1}}\exp(-Q/\mathrm{RT})\left(\alpha \sigma <0.8\right)$$
(7)$$\dot{\varepsilon }=A_{2}\exp(\beta \sigma )\exp(-Q/\mathrm{RT})\left(\alpha \sigma >1.2\right)$$
(8)$$\dot{\varepsilon }=A{\left[\sin h(\alpha \sigma )\right]}^{n}\exp(-Q/\mathrm{RT})(\mathrm{for}\;\mathrm{all}\sigma )$$
where $\dot{\varepsilon }$, *Q*, *R*, *T*, *σ* are strain rate (s^−1^), thermal deformation activation energy (J/mol), molar gas constant (8.314 J/(mol · K)), deformation temperature (*K*), and flow stress, respectively. A,A_1_, A_2_, *n*_1_, *n*, β, and α (α = β/*n*_1_) are material constants. Table 3 shows the peak stresses of TA32 titanium alloy at varying deformation parameters.(9)$$\ln\dot{\varepsilon }=\ln A_{1}+n_{1}\ln\sigma_{p}-Q/\mathrm{RT}$$(10)$$\ln\dot{\varepsilon }=\ln A_{2}+{\beta \sigma }_{p}-Q/\mathrm{RT}$$(11)$$\ln\dot{\varepsilon }=\ln A-Q/\mathrm{RT}+n\ln\left[\sin h(\alpha \sigma_{p})\right]$$

Equation (9) to Equation (11) are functional relationships between strain rates and stress obtained by taking logarithms on both sides. Bringing in the peak stress and deformation rates at the corresponding temperatures, a linear fit to lnσ − $\ln\dot{\varepsilon }$ and σ − $\ln\dot{\varepsilon }$ yields a stress—strain rate relationship curve, as shown in Figure 6a,b. The linear regression equations are obtained by calculating the slopes corresponding to the $n_{1}$ and β values. The average values of β and *n*_1_ were 0.037 and 11.18, and the value of α was 0.0033. Figure 6c describes the relationship of $\ln\dot{\varepsilon }$ − $\ln\left[\sin h(\alpha \sigma_{p})\right],$ and it is obvious that the linear fit correlation is high at all temperatures. The value of the stress exponent *n* is shown by the slope of the straight line under different deformation temperatures. Averaging the slopes yields the stress exponent *n* of 7.37 in the dual-phase zone. When the strain rates are certain, assuming that the magnitude of the thermal activation energy *Q* value is independent of temperature, the following equation can be introduced:(12)$$\frac{Q}{nR}={\left.\frac{\partial \ln\left[\sin h(\alpha \sigma_{p})\right]}{\partial (1/T)}\right|}_{\dot{\varepsilon }}$$

The functional relationship of $\ln\left[\sin h(\alpha \sigma_{p})\right]$ − 1000/*T* can be acquired according to Equation (12). The value of the slope with the linear fit is *Q*/1000*nR*, as seen in Figure 6d. It was calculated that the average slope of the curve was 6.83 for the temperature interval from 700 to 950 °C. To calculate the activation energy, substitute *R* and the average stress exponent value n into Equation (12), which is calculated to be *Q* = 418.51 kJ/mol. The peak stress of TA32 titanium alloy is linearly fitted to lnZ − $\ln\left[\sin h(\alpha \sigma_{p})\right]$, and the final value of *A* is 1.09 × 10^25^.

Substituting the calculated values of *Q*, *n*, *A*, and α, the prediction model for peak stress in the dual-phase zone of TA32 titanium alloy can be obtained as follows:(13)$$\left\{\begin{array}{c} \dot{\varepsilon }=1.57\times 10^{41}{\left[\sin h\left(0.0024\sigma \right)\right]}^{13.45}\exp(-642984/RT)\\ \sigma =(1/0.0024)\ln\left\{{\left(\frac{Z}{1.57\times 10^{41}}\right)}^{1/13.45}+{\left[{\left(\frac{Z}{1.57\times 10^{41}}\right)}^{2/13.45}+1\right]}^{1/2}\right\}\\ Z=\dot{\varepsilon }\exp(642984/RT) \end{array}\right.$$

To assess the precision of the TA32 alloy peak stress prediction model, the precision is characterized by two parameters: the absolute value of the average relative error (*ARRE*) and the correlation coefficient (*R*) [30], with the following expressions:(14)$$R=\frac{\sum_{i=1}^{N}\left(E_{i}-\bar{E})(P_{i}-\bar{P}\right)}{\sqrt{\sum_{i=1}^{N}{\left(E_{i}-\bar{E}\right)}^{2}{\sum_{i=1}^{N}\left(P_{i}-\bar{P}\right)}^{2}}}$$(15)$$\mathrm{ARRE}=\frac{1}{N}\sum_{i=1}^{N}\left|\frac{E_{i}-P_{i}}{E_{i}}\right|$$
where *P* is the predicted data from the intrinsic model, *E* is the number of corrected peak stress data, *N* is the total number of peak stresses, and $\bar{P},\bar{E}$ are the average values of *P* and *E*, respectively. Substituting the peak stress values corrected by friction and temperature and the predicted peak stress data of TA32 titanium alloy into Equation (14) to Equation (15), the degree of deviation between the experimental values and the predicted values can be acquired, as demonstrated in Figure 7. It is obvious that the correlation coefficient and the average relative error of TA32 titanium alloy are 95.1% and 10.9%, respectively. The results describe that peak stress prediction can precisely forecast peak stress under deformation conditions from 700 to 950 °C and 0.001 to 1.0 s^−1^.

### 3.3. Microstructure Evolution

#### 3.3.1. Effect of Strain Rate on Microstructure

Figure 8a–c show the microstructure during deformation at 950 °C and a strain rate of 0.01 to 1.0 s^−1^. At a 0.01 s^−1^ strain rate, the microstructure is made up of primary equiaxed α and short rod-shaped α, as can be seen. The lamellar α is no longer observed in the microstructure at a 1.0 s^−1^ strain rate, and numerous acicular α exist, which are precipitated from the β phase [31,32]. Figure 8d displays the results of the average size and volume percentage of primary equiaxed α at various strain rates. The amount of primary equiaxed α steadily dropped as temperatures rose, while the size of equiaxed α showed no significant change. This is because under this deformation condition, the adiabatic temperature rise effect is not prominent, and no obvious phase transition occurs, so the content of primary α is similar to that of the original microstructure. In addition, at low deformation rates, the fragmentation of lamellar α is high, and a large number of short rod-shaped α are present in the microstructure. The effect of the adiabatic temperature rise is substantial at a 1.0 s^−1^ strain rate. According to the previous calculation of temperature rise, the value of temperature rise under this condition is close to 20 °C, so the actual temperature in the central region is about 970 °C. When the microstructure undergoes a phase transition, the lamellar α_s_ preferentially transforms into the β compared to the equiaxed α_p_. Under this deformation condition, the secondary lamellar α in the microstructure has already completely transformed into the β due to the phase transition before deformation, and some equiaxed α have also undergone the phase transition [33,34,35]. Therefore, the transformed β in phase transition re-precipitates as acicular α during the cooling process, and the volume percentage of the equiaxed α drops to 34.1%.

Figure 9 describes the grain boundary orientation difference distribution diagram and the grain boundary diagram of TA32 alloy at 950 °C and 0.01 to 1.0 s^−1^ strain rates. Low angle grain boundaries (LAGBs) are represented by red lines with an angle between 2° and 15° in the GB diagram, while high angle grain boundaries (HAGBs) are represented by black lines with an angle of more than 15°. The impact of strain rate on subgrain, misorientation angle distribution, and grain morphology is significant. The proportions of HAGBs and LAGBs were 63.2% and 36.8% at a 0.01 s^−1^ strain rate, as shown in Figure 9d–f, and the peaks in the 2° to 5° range became very prominent. However, as the deformation rates rose to 0.1 s^−1^ and 1.0 s^−1^, the proportion of LAGBs decreased to 32.3% and 17.7%, significantly lower than 0.01 s^−1^. It is evident that the driving force for the conversion from LAGBs to HAGBs rises as the strain rate increases [36].

Figure 10a–f depicts the KAM and GOS diagrams of TA32 titanium alloy at 950 °C deformation temperature with varying deformation rates. It is obvious that above a certain temperature, the dislocation tangle in the microstructure weakens, and the KAM value decreases as strain rates increase. This is because the deformation heat generated by the adiabatic temperature rise effect does not have time to disappear at high strain rates. The temperature rise provides power for the atomic motion, and the driving force for dislocation annihilation is strengthened, causing the KAM distribution to decrease as the strain rates increase. At a 0.01 s^−1^ strain rate, many LAGBs appear in the microstructure, and dislocations are concentrated in a few grains, resulting in a strengthened dislocation block and higher KAM values. On the other hand, Figure 9 also shows that as deformation rates rise, the volume proportion of HAGBs and subgrain structures increases dramatically, leading to the absorption of dislocations and a decrease in the KAM value [37,38,39]. As shown in Figure 10g, the KAM distribution diagram, with an increase in strain rate, the average orientation difference angle of KAM reduces from 0.88° to 0.82° and finally decreases to 0.58°. The dislocation density also significantly decreases.

Recrystallized grains are those whose GOS is smaller than 2° in the microstructure, and Figure 10h shows the changes in the volume fraction and average grain size of DRX as deformation rates rise. The volume fraction and average grain size of DRX decreased from 34.6% and 3.03 μm at a 0.01 s^−1^ strain rate to 28.1% and 2.58 μm at a 0.1 s^−1^ strain rate and then to 11.3% and 2.31 μm at a 1.0 s^−1^ strain rate, according to the statistical values. It shows that with an increase in strain rates, the volume fraction of recrystallization likewise decreases, and the size of recrystallized grains reduces to varying degrees. This is because longer deformation times encourage the occurrence of DRX under low deformation rates, and DRX grains take more time to develop after nucleation [40]. In contrast, at a 1.0 s^−1^ strain rate, due to the non-instantaneous growth time and short thermal deformation time, it occurs with a lower DRX grain size. In addition, under this deformation condition, the acicular α precipitate also inhibits the nucleation of DRX, and the deformation mechanism transforms from DRX to dynamic recovery (DRV) [41,42].

#### 3.3.2. Effect of Deformation Temperature on Microstructure

Figure 11a–c describe the microstructure after deformation at a 0.1 s^−1^ strain rate at 800 to 900 °C. At low deformation temperatures in the dual-phase zone, there was no significant phase transition in the microstructure. Therefore, the content of primary equiaxed α was nearly unchanged, and the main differences were reflected in the average size of α_p_ grains. Figure 11 shows that the primary equiaxed α grains undergo an evident elongation along the vertical compression direction (CD) at 850 °C temperature compared to the original microstructure. At 900 °C deformation temperature, the high deformation temperatures promote the grain boundary diffusion and increase the equiaxed degree of α_p_ in the microstructure, resulting in a more uniform morphology [43]. The average grain size at varying deformation parameters is displayed in Figure 11d. The results indicate that the average grain size rises from 11.65 μm at 800 °C to 12.23 μm at 850 °C as the deformation temperature increases. Finally, the grain size rises significantly to 14.13 μm at 900 °C.

Figure 12 indicates the grain boundary orientation difference distribution diagrams and grain boundary diagrams of TA32 alloy under the conditions of 800 to 900 °C. Figure 12a–c show that the percentage of HAGBs rises as deformation temperatures increase, while the percentage of LAGBs reduces significantly. In Figure 12d–f, quantitative statistics were conducted on the percentage of HAGBs and LAGBs. At an 800 °C deformation temperature, the content of HAGBs and LAGBs was 67.1% and 32.9%, respectively. However, with deformation temperatures rising, the proportion of LAGBs reduced to 58.9% and 52.1% at 850 and 900 °C, respectively. This is because at a low deformation temperature, the driving force of dynamic recrystallization is insufficient, and numerous subgrain structures and LAGBs are generated in deformed grains, causing a low degree of DRX. The nucleation and development of recrystallized grains are promoted by a rise in deformation temperature, and DRX grains continuously consume LAGBs and convert them into HAGBs in the growth process [44,45]. The increase and migration of HAGBs with increasing temperature during thermal deformation demonstrates that DRX progressively becomes the predominant softening mechanism.

Figure 13a–f show the KAM and GOS diagrams of TA32 titanium alloy at 0.1 s^−1^ strain rate and deformation temperatures ranging from 800 to 900 °C. Figure 13g presents a statistical analysis of the KAM values under three temperature conditions. The results describe that the dislocations are more likely to slide, and the KAM values reduce as deformation temperatures increase. It is obvious that the distribution location of KAM is extremely analogous to LAGBs, so analysis should be conducted from the perspective of recrystallization. In Figure 13h, a quantitative analysis was conducted on the average grain size and volume fraction of recrystallization. It is obvious that a rise in temperatures promotes the incidence of recrystallization, and higher deformation temperatures provide kinetic energy for atoms in the microstructure, promoting the development and nucleation of recrystallization grains. The average grain size and volume proportion of recrystallization increased from 22.1% and 1.54 μm at 800 °C to 27.6% and 1.56 μm at 850 °C and finally increased to 28.4% and 1.71 μm at 900 °C. The increase in DRX degree consumes numerous dislocations, and DRX grains are mainly distributed around the deformed crystal because the deformed region has higher dislocation density and energy, making it the preferred location for grain nucleation and further reducing dislocations [46,47].

### 3.4. Analysis of DRX Mechanisms

DRX always nucleates preferentially at the more deformed areas during plastic deformation, with two main modes—CDRX and DDRX. The two modes can be distinguished by whether the cumulative orientation difference within the grain is greater than 15° [48]. The IPF map with the angular distribution of the orientation difference along the L_1_ direction at 0.001 s^−1^ and 900 °C condition is given in Figure 14. It is evident that a distinct color gradient is present inside the deformed grains, illustrating that the deformation within the grains is not homogeneous, which leads to the change in lattice orientation. According to Figure 14a, the equiaxed α_p_ grains are subdivided into three subcrystalline regions by small-angle grain boundaries and labeled as 1, 2, and 3. Figure 14b shows that the cumulative orientation difference (pointing to the origin) along the L_1_ direction steadily increases, and the final cumulative orientation difference exceeds 15°. Meanwhile, the point-to-point orientation difference angles have a sudden change at the subcrystalline boundary, which shows an obvious “peak” characteristic. The increase in the orientation difference between neighboring subgranules shows that the gradual rotation process of subgranules is active, which is a characteristic feature of CDRX, which starts with the generation and proliferation of dislocations. As strain increases, the dislocations are rearranged and transformed into new dynamically recrystallized grains by gradual rotation of the subgrains. The formation of substructures with LAGBs inside the deformed grains usually induces dynamic recrystallization as the grain boundary angle increases, showing a typical CDRX mechanism [49]. Figure 14c describes the appearance of the gradual rotation of the subgrains. Numerous dislocations migrate to the surrounding subcrystalline boundaries by creep and slip, leading to the removal of the initial subcrystalline boundaries. Through the successive rotations of the subgrains, the LAGBs are gradually dissipated and absorbed, eventually transforming into the HAGBs [50,51].

Figure 15 demonstrates the DDRX mechanism of TA32 alloy at 0.1 s^−1^ and 800 °C deformation conditions. It is evident that the point-spot orientation difference angle is basically less than 2°, and the accumulated point-source orientation difference angle at the grain boundaries is less than 7°, indicating the formation of DDRX grains [52]. DDRX is typically characterized by the strain-induced grain boundary migration, which would occur when DDRX grains nucleate along the grain boundaries and form a jagged grain boundary by grain boundary migration. The undeformed locations generate protrusions behind the jagged grain boundaries, which ultimately evolve into nuclei. These nuclei eventually develop into new DRX grains containing HAGBs [53].

## 4. Conclusions

In this paper, the peak stress prediction model of TA32 titanium alloy, considering friction-temperature correction, is established. The influence of deformation parameters on microstructure evolution during thermal deformation is comprehensively analyzed based on the EBSD characterization. These findings provide reference and guidance for the microstructure evolution and thermal deformation behavior of TA32 titanium alloy in the biphase region. The study is helpful for the subsequent research of thermal deformation behavior and the establishment of a constitutive model in the single-phase region. There are differences between the setting of experimental conditions and the environment of specific production practice, and the subsequent experimental scheme design can be further optimized to be more in line with the thermal process route of actual processing and production so as to promote the practical application and promotion of TA32 titanium alloy. The primary conclusions are as follows:The flow stress rises quickly to the peak value, and then gradually reduces and eventually stabilizes under the softening impact of DRX at the beginning of deformation.The experimental values are significantly lower than the friction-temperature-corrected stress values, and a prediction model for the peak stress of TA32 alloy in the dual-phase zone has been established based on the correction results, as follows:$$\left\{\begin{array}{c} \dot{\varepsilon }=1.57\times 10^{41}{\left[\sin h\left(0.0024\sigma \right)\right]}^{13.45}\exp(-642984/\mathrm{RT})\\ \sigma =(1/0.0024)\ln\left\{{\left(\frac{Z}{1.57\times 10^{41}}\right)}^{1/13.45}+{\left[{\left(\frac{Z}{1.57\times 10^{41}}\right)}^{2/13.45}+1\right]}^{1/2}\right\}\\ Z=\dot{\varepsilon }\exp(642984/\mathrm{RT}) \end{array}\right.$$At 950 °C temperature deformation, a high strain rate (1.0 s^−1^) under the adiabatic temperature rise effect is obvious. This condition precipitated a large number of acicular α, which inhibited the nucleation and development of DRX. The main softening mechanism shifted to DRV.DRX is the primary mechanism during the thermal deformation process, and the volume fraction and average size of DRX grains rise as temperatures increase. The recrystallization mode is dominated by CDRX with increasing temperatures and decreasing strain rates. At higher strain rates and lower temperatures, the DDRX mechanism exists.

## Figures and Tables

**Figure 1 materials-18-01476-f001:**
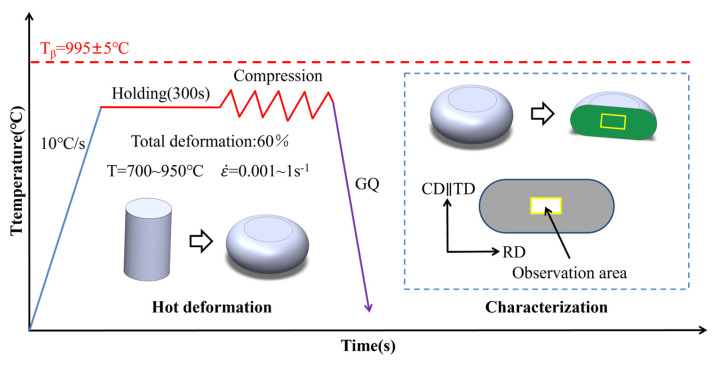
TA32 alloy hot compression process route (GQ: Gas-Quenching).

**Figure 2 materials-18-01476-f002:**
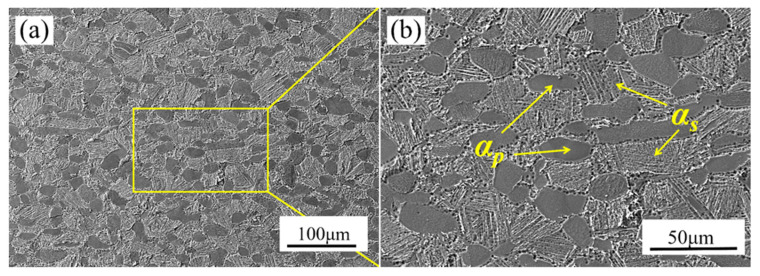
SEM image of initial microstructure of TA32 alloy: (a) 500× and (b) 1000×.

**Figure 3 materials-18-01476-f003:**
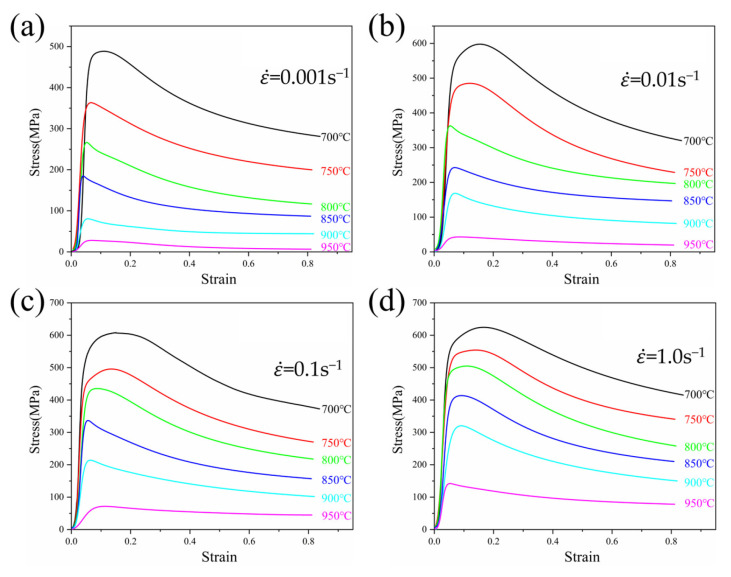
Stress–strain curves of TA32 alloy under different deformation conditions: (**a**) 0.001 s^−1^; (**b**) 0.01 s^−1^; (**c**) 0.1 s^−1^; (**d**) 1.0 s^−1^.

**Figure 4 materials-18-01476-f004:**
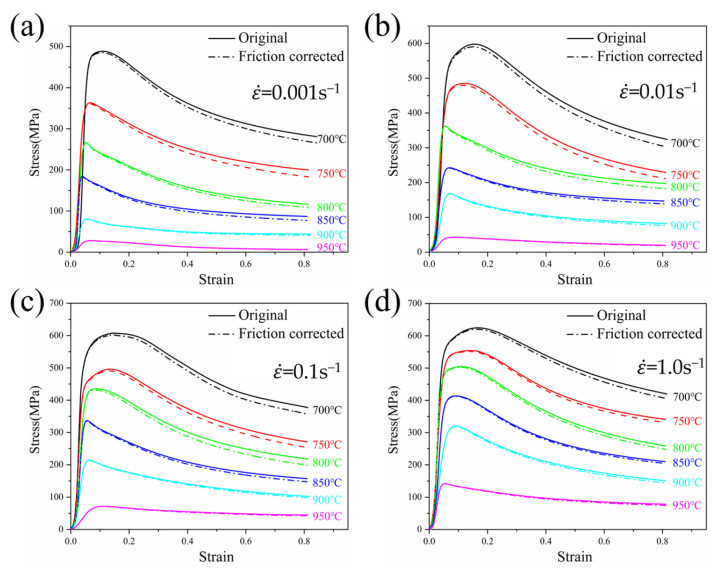
Stress–strain curve of TA32 titanium alloy after friction correction: (**a**) 0.001 s^−1^; (**b**) 0.01 s^−1^; (**c**) 0.1 s^−1^; (**d**) 1.0 s^−1^.

**Figure 5 materials-18-01476-f005:**
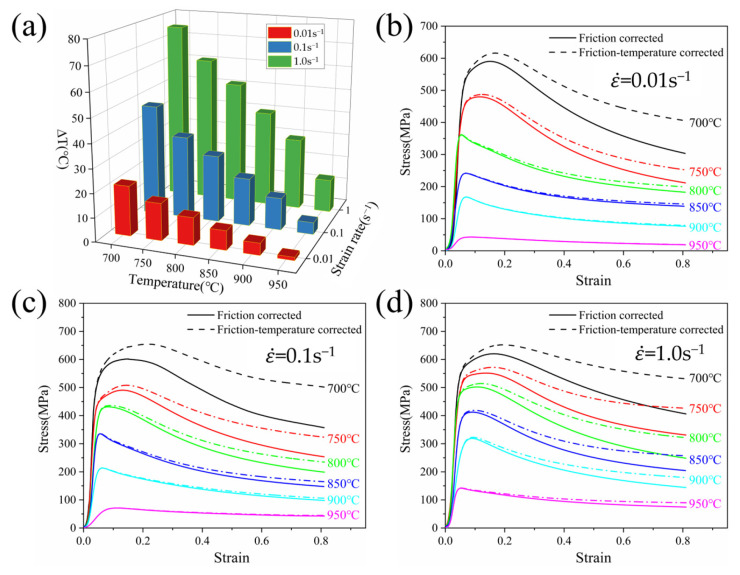
The friction-temperature correction stress–strain curves of TA32 alloy: (**a**) value of temperature change; (**b**) 0.01 s^−1^; (**c**) 0.1 s^−1^; (**d**) 1.0 s^−1^.

**Figure 6 materials-18-01476-f006:**
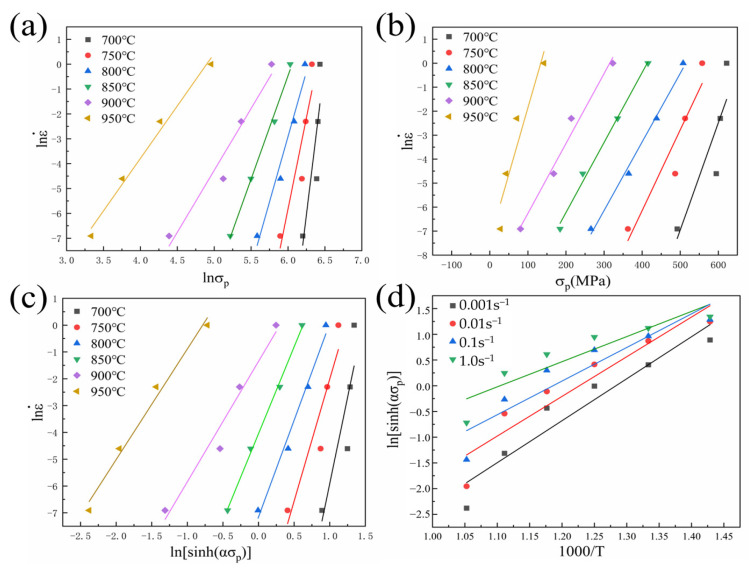
Functional relationship of the principal constitutive equation: (**a**) $\ln\sigma_{p}$ − $\ln\dot{\varepsilon }$; (**b**) $\sigma_{p}$ − $\ln\dot{\varepsilon }$; (**c**) $\ln\dot{\varepsilon }-\ln\left[\sin h(\alpha \sigma_{p})\right]$; (**d**) $\ln\left[\sin h(\alpha \sigma_{p})\right]-1000/T$.

**Figure 7 materials-18-01476-f007:**
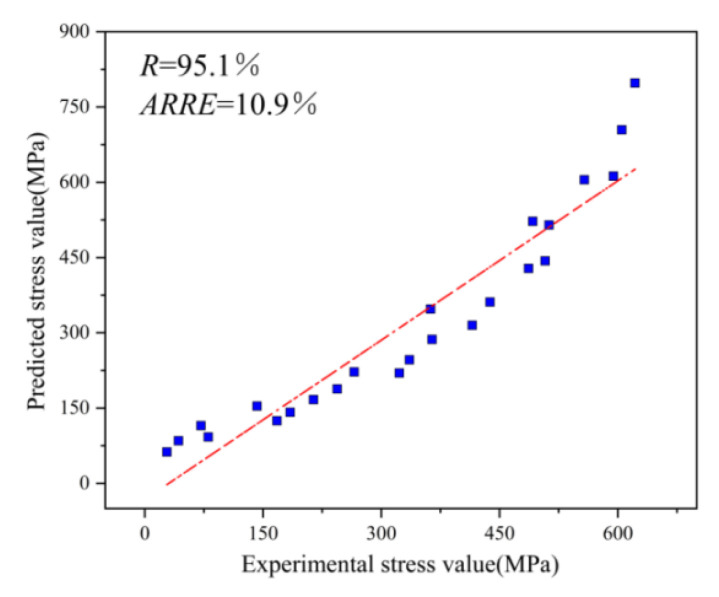
Correlation graph between predicted and experimental peak stress values of TA32 alloy.

**Figure 8 materials-18-01476-f008:**
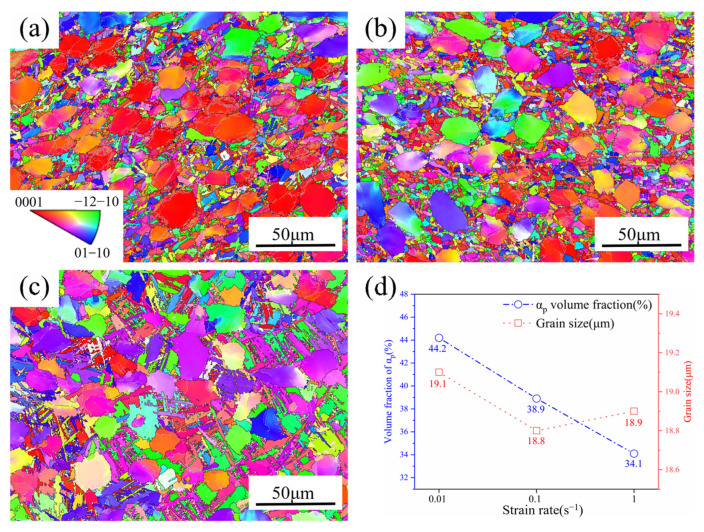
The IPF distribution maps of TA32 alloy at 950 °C: (**a**) 0.01 s^−1^; (**b**) 0.1 s^−1^; (**c**) 1.0 s^−1^; (**d**) equiaxial α content and grain size distribution.

**Figure 9 materials-18-01476-f009:**
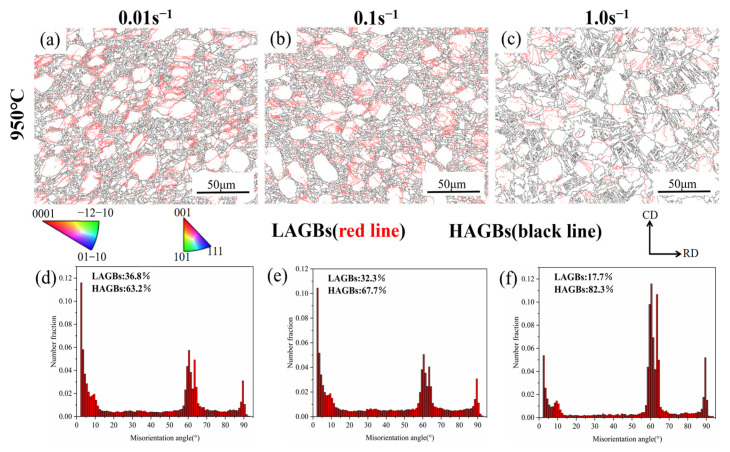
The grain boundary maps of TA32 alloy at 950 °C: (**a**,**d**) 0.01 s^−1^; (**b**,**e**) 0.1 s^−1^; (**c**,**f**) 1.0 s^−1^; (**d**–**f**) distribution diagram of grain boundary orientation difference.

**Figure 10 materials-18-01476-f010:**
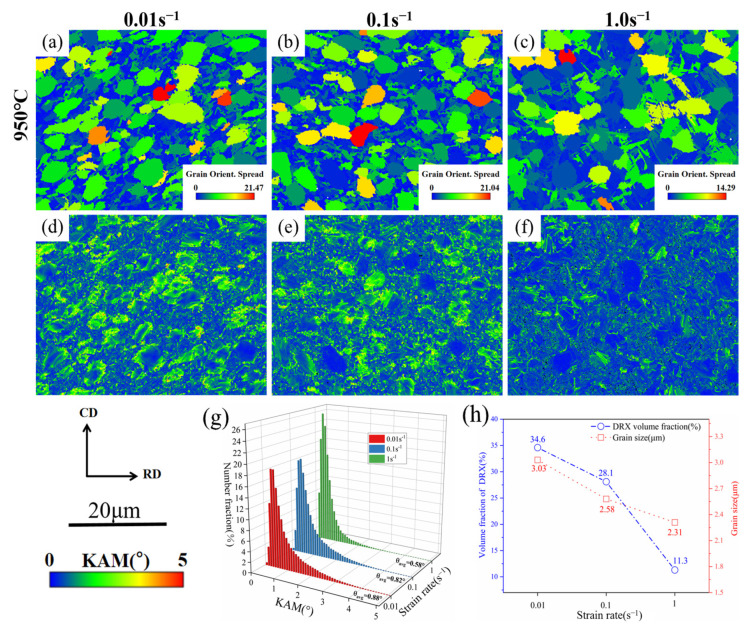
The grain orientation spread (GOS) diagram (**a**–**c**) and the KAM diagram (**d**–**f**) at 950 °C deformation temperature under different strain rates for (**a**,**d**) 0.01 s^−1^, (**b**,**e**) 0.1 s^−1^, and (**c**,**f**) 1.0 s^−1^; (**g**) KAM distribution; (**h**) DRX average grain size and volume fraction distribution diagram.

**Figure 11 materials-18-01476-f011:**
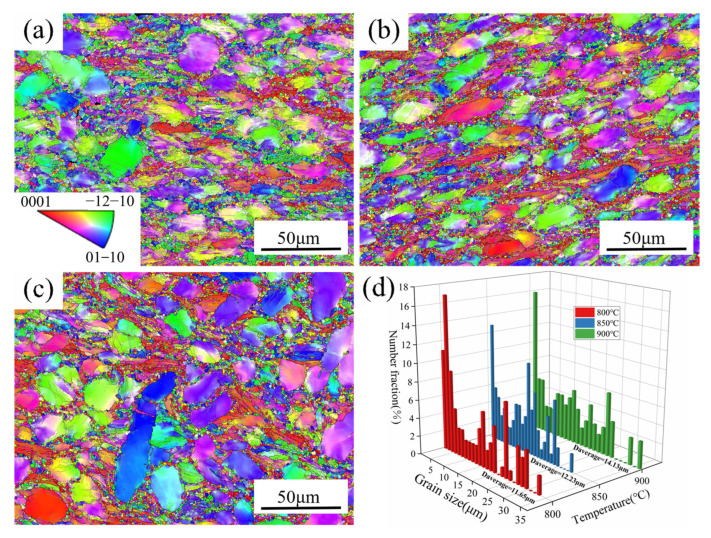
The IPF distribution maps of TA32 alloy at 0.1 s^−1^: (**a**) 800 °C, (**b**) 850 °C, and (**c**) 900 °C; (**d**) distribution of average grain size.

**Figure 12 materials-18-01476-f012:**
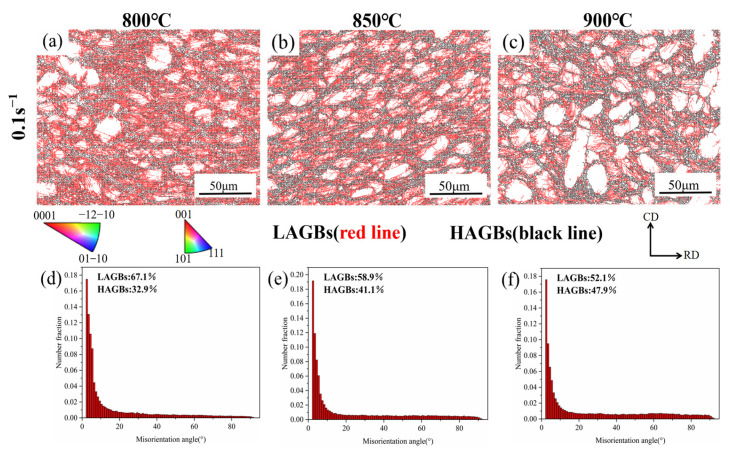
The grain boundary diagrams of TA32 alloy at 0.1 s^−1^: (**a**,**d**) 800 °C, (**b**,**e**) 850 °C, and (**c**,**f**) 900 °C; (**d**–**f**) distribution diagrams of grain boundary orientation difference.

**Figure 13 materials-18-01476-f013:**
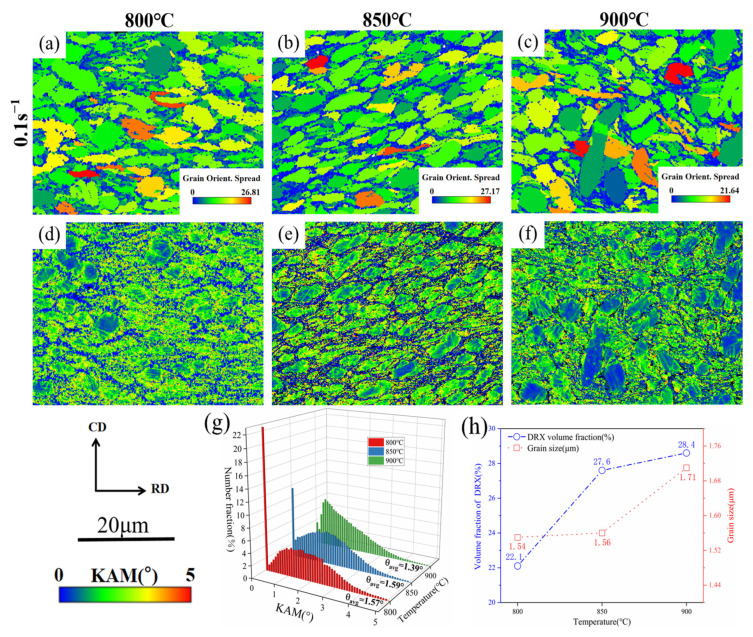
The grain orientation spread (GOS) diagrams (**a**–**c**) and the KAM diagrams (**d**–**f**) at 0.1 s^−1^ strain rate under varying temperatures for (**a**,**d**) 800 °C, (**b**,**e**) 850 °C, and (**c**,**f**) 900 °C; (**g**) KAM distribution; (**h**) DRX volume fraction and average grain size distribution diagram.

**Figure 14 materials-18-01476-f014:**
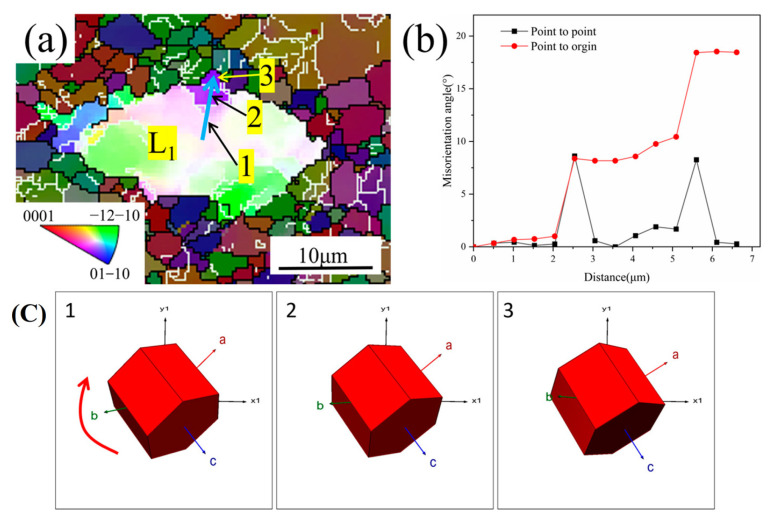
CDRX mechanism of TA32 alloy at 900 °C and 0.001 s^−1^: (**a**) IPF map; (**b**) distribution of orientation difference angle along L_1_ direction; (**c**) changes in grain orientation.

**Figure 15 materials-18-01476-f015:**
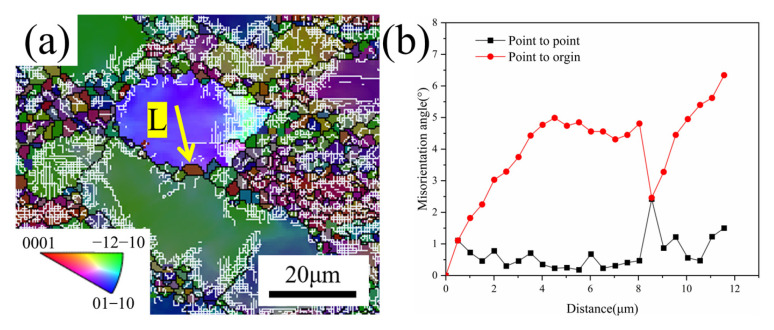
DDRX mechanism of TA32 alloy at 800 °C and 0.1 s^−1^ deformation: (**a**) IPF map; (**b**) distribution of orientation difference angle along the L direction.

**Table 1 materials-18-01476-t001:** Chemical composition of TA32 alloy (wt %).

Element	Al	Sn	Zr	Mo	Nb	Ta	Si	Ti
Content	5.5	3.5	3.0	1	0.5	0.7	0.3	85.5

**Table 2 materials-18-01476-t002:** The bulging coefficient *B* under different deformation conditions.

Strain Rate (s^−1^)	Temperature of Deformation (°C)					
	700	750	800	850	900	950
0.001	1.249	1.371	1.336	1.463	1.306	1.117
0.01	1.253	1.275	1.304	1.367	1.330	1.197
0.1	1.255	1.217	1.313	1.311	1.319	1.376
1	1.065	1.066	1.203	1.232	1.368	1.258

**Table 3 materials-18-01476-t003:** Peak stresses of TA32 alloy under different deformation parameters (MPa).

Strain Rate (s^−1^)	Temperature of Deformation (°C)					
	700	750	800	850	900	950
0.001	492.02	362.55	265.67	184.52	80.58	28.04
0.01	594.32	486.67	364.47	244.06	167.72	42.81
0.1	604.95	512.75	437.89	335.61	214.02	71.36
1	621.82	557.51	507.68	415.34	322.86	142.45

## Data Availability

The original contributions presented in the study are included in the article. Further inquiries can be directed to the corresponding author.

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
