# Peer review of "Hot Deformation Behavior and Microstructure Evolution of Near-α Titanium Alloy TA32 in Dual-Phase Zone"

_materials, 2025, doi:10.3390/ma18071476_

Round 1

Reviewer 1 Report

Comments and Suggestions for Authors

The authors studied about the hot deformation behavior with dynamic recrystalization in the dual phase zone of near-α titanium alloy TA32.

The experiments in the wide variety of temperature and strain rate are carried out and the constitutive models based on the thermal activation as well as microstructural observations by EBSD are discussed.

Unfortunately, there are some unclear points. I do not recommend this paper in the present form for the submission to Materials.

The following comments are available.

1. The model is so phenomenological and frequently-used. I do would like to ask for the authors what is new on the model. A use of different material provides the different parameters in the model. This is common.

2. Related to the above, each parameter in the models is not connected to the microstructural features as shown in the later part of the manuscript. 

3. On Fig.4, how to compensate  the effect of friction? Detailed procedures should be written. How to determine the friction coefficient?

4. What kind of testing machine is used? Greeble?

Author Response

Thank you for your valuable comments and suggestions on our manuscript titled "Hot deformation behavior and microstructure evolution of near-α titanium alloy TA32 in dual-phase zone" . We have carefully revised the manuscript according to the reviewers' comments. Below, we provide a point-by-point response to the reviewers' concerns.

Comments 1: The model is so phenomenological and frequently-used. I do would like to ask for the authors what is new on the model. A use of different material provides the different parameters in the model. This is common.

Response 1: Thank you for pointing this out. We agree with this comment. Based on the TA32 biphase titanium alloy used in this paper, the relevant parameters used to establish this constitutive model are given in detail, which are different from other materials, including the density, specific heat capacity, and the selection of adiabatic correction coefficient of this titanium alloy. This change can be found on page 7, paragraph 1, lines 219-222.

Comments 2: Related to the above, each parameter in the models is not connected to the microstructural features as shown in the later part of the manuscript. 

Response 2: Thank you for pointing this out. We agree with this comment. The parameters in the model can reflect the structural characteristics of the microstructure. In the introduction we have added a paragraph on the properties of the mathematical parameters of the constitutive model and their intrinsic relation to the microstructure. These detailed changes can be found on page 2, paragraph 2, lines 51-69.

Comments 3: On Fig.4, how to compensate  the effect of friction? Detailed procedures should be written. How to determine the friction coefficient?

Response 3: Thank you for pointing this out. We agree with this comment. We added the detailed process of friction correction and how to determine the drum coefficient and friction coefficient. This information can be found on pages 5-6.

Comments 4: What kind of testing machine is used? Greeble?

Response 4: Thank you for pointing this out. We agree with this comment. The thermal compression experiment of TA32 titanium alloy was carried out by Thermecamastor-z thermal simulation tester. These detailed changes can be found on page 3, paragraph 1, lines 103-104.

We hope that the revised manuscript meets the journal's standards. Please let us know if further modifications are needed.

Sincerely,
Authors

Reviewer 2 Report

Comments and Suggestions for Authors

 The manuscript “Hot deformation behavior and microstructure evolution of near-α titanium alloy TA32 in dual-phase zone” is a study relevant. However, before publishing this study, I suggest that an extensive revision of the English language is needed. There are paragraphs where it is difficult to understand what the authors are trying to explain.

I also suggest adding a section after the introduction describing the characteristics of the mathematical model that was used to predict the mechanical properties.

Comments on the Quality of English Language

 The manuscript “Hot deformation behavior and microstructure evolution of near-α titanium alloy TA32 in dual-phase zone” is a study relevant. However, before publishing this study, I suggest that an extensive revision of the English language is needed. There are paragraphs where it is difficult to understand what the authors are trying to explain.

Author Response

Thank you for your valuable comments and suggestions on our manuscript titled "Hot deformation behavior and microstructure evolution of near-α titanium alloy TA32 in dual-phase zone" . We have carefully revised the manuscript according to the reviewers' comments. Below, we provide a point-by-point response to the reviewers' concerns.

Comments 1: The manuscript “Hot deformation behavior and microstructure evolution of near-α titanium alloy TA32 in dual-phase zone” is a study relevant. However, before publishing this study, I suggest that an extensive revision of the English language is needed. There are paragraphs where it is difficult to understand what the authors are trying to explain.

Response 1: Thank you for pointing this out. We agree with this comment. We have made extensive corrections to the English expression of the paper, especially the experimental part to optimize the expression to make it easier to understand.

Comments 2: I also suggest adding a section after the introduction describing the characteristics of the mathematical model that was used to predict the mechanical properties.

Response 2:Thank you for pointing this out. We agree with this comment. In the introduction we have added a paragraph on the properties of the mathematical parameters of the constitutive model and their intrinsic relation to the microstructure. These detailed changes can be found on page 2, paragraph 2, lines 51-69.

We hope that the revised manuscript meets the journal's standards. Please let us know if further modifications are needed.

Sincerely,
Authors

Reviewer 3 Report

Comments and Suggestions for Authors

The manuscript is based on a study of the hot deformation behavior and microstructure evolution of near-α titanium alloy TA32 in a dual-phase zone. By using the isothermal compression experiments of a model of TA32 titanium alloy, the findings provide certain reference and theoretical value for the hot deformation process of TA32 alloy. In my opinion, the manuscript needs revisions and should be more organized, and authors should improve focus.

Comments

  1. The manuscript should be more organized, and authors should improve focus, especially by clearly defining the main objective of the study: abstract and introduction.
  2. In case of range, the first unity should be removed. Examples: 700 to 950 ℃… 001 to 1.0 s⁻¹…. etc.
  3. Table 1 and Fig. 1 should be presented in the ‘results’ section.
  4. Experimental procedures should be presented as sub-sections. Examples: thermal analysis, mechanics/rheology, SEM analysis, distribution maps, etc.
  5. The graphical quality of the main figures should be improved for more visibility.
  6. The conclusion section should be revised. The authors should promote “conclusions” in terms of including limitations of this study, as well as highlight the implications and perspectives of the study.
Comments on the Quality of English Language

I am not qualified to assess the quality of English in this paper. But generally, I think the English could be improved to more clearly express the research.

Author Response

Thank you for your valuable comments and suggestions on our manuscript titled "Hot deformation behavior and microstructure evolution of near-α titanium alloy TA32 in dual-phase zone" . We have carefully revised the manuscript according to the reviewers' comments. Below, we provide a point-by-point response to the reviewers' concerns.

Comments 1: The manuscript should be more organized, and authors should improve focus, especially by clearly defining the main objective of the study: abstract and introduction.

Response 1: Thank you for pointing this out. We agree with this comment. We have revised the abstract and introduction to highlight the key points, clarify the research objectives and contents, especially explain the parameter characteristics of the established constitutive model and describe its internal relationship with the microstructure. This change can be seen on page 2, paragraph 2, lines 51-69 and in the abstract section.

Comments 2: In case of range, the first unity should be removed. Examples: 700 to 950 ℃… 001 to 1.0 s⁻¹…. etc.

Response 2: Thank you for pointing this out. We agree with this comment. In this paper, the first unit is deleted from the range of temperature or speed.

Comments 3: Table 1 and Fig. 1 should be presented in the ‘results’ section.

Response 3: Thank you for pointing this out. We agree with this comment. Table 1 is the chemical composition provided by the supplier, so it is put in the part of material preparation before the hot compression test. In the result part, the initial structure of TA32 titanium alloy is analyzed and explained. This change can be seen on page 4, paragraph 1.

Comments 4: Experimental procedures should be presented as sub-sections. Examples: thermal analysis, mechanics/rheology, SEM analysis, distribution maps, etc.

Response 4: Thank you for pointing this out. We agree with this comment. The experimental procedure is no longer a general description, but is specifically divided into material preparation, thermal deformation experiment, tissue characterization and other parts. This change can be seen on page 3.

Comments 5: The graphical quality of the main figures should be improved for more visibility.

Response 5: Thank you for pointing this out. We agree with this comment. We reworked some of the images in the paper to improve their visibility.

Comments 6: The conclusion section should be revised. The authors should promote “conclusions” in terms of including limitations of this study, as well as highlight the implications and perspectives of the study.

Response 6: Thank you for pointing this out. We agree with this comment. We modified the conclusion part, and put forward the possibility and guiding significance of follow-up research on the limitations of the study, in order to promote the application of this titanium alloy in engineering production. This change can be seen on page 17, paragraphs 2-3.

Sincerely,
Authors

Reviewer 4 Report

Comments and Suggestions for Authors
  1. The manuscript in the Materials section does not specify the source or supplier of the TA32 alloy.
  2. In Table 1, the source of the information is not indicated. For example, it would be helpful to specify that the chemical composition of the TA32 alloy (wt%) was provided by the supplier.
  3. The resolution of the figures should be improved, as in several of them the legends are illegible. 7, 8, 9, 10, 11, and 12.
  4. The authors have written the full name of the journal, but the journal requires its abbreviation.

References should be described as follows, depending on the type of work:

  • Journal Articles:
    1. Author 1, A.B.; Author 2, C.D. Title of the article. Abbreviated Journal Name YearVolume, page range.

Author Response

Thank you for your valuable comments and suggestions on our manuscript titled "Hot deformation behavior and microstructure evolution of near-α titanium alloy TA32 in dual-phase zone" . We have carefully revised the manuscript according to the reviewers' comments. Below, we provide a point-by-point response to the reviewers' concerns.

Comments 1: The manuscript in the Materials section does not specify the source or supplier of the TA32 alloy.

Response 1: Thank you for pointing this out. We agree with this comment. In this paper, we have specified the supplier of the raw materials selected in this paper. This change can be seen on page 2, lines 96-97.

Comments 2: In Table 1, the source of the information is not indicated. For example, it would be helpful to specify that the chemical composition of the TA32 alloy (wt%) was provided by the supplier.

Response 2: Thank you for pointing this out. We agree with this comment. We have stated in this paper that the source of the chemical composition of the raw materials selected in this paper is provided by the supplier. This change can be seen on page 3, lines 100-101.

Comments 3: The resolution of the figures should be improved, as in several of them the legends are illegible. 7, 8, 9, 10, 11, and 12.

Response 3: Thank you for pointing this out. We agree with this comment. All figures are in bold font to make them easier to read.

Comments 4: The authors have written the full name of the journal, but the journal requires its abbreviation.

Response 4: Thank you for pointing this out. We agree with this comment. We have made extensive revisions to the format of the references required by the journal.

Sincerely,
Authors

Round 2

Reviewer 1 Report

Comments and Suggestions for Authors

Dear the authors,

Thank you very much for the authors' great efforts.

Now, I would like to recommend this manuscript for the publication to Materials.

Reviewer 2 Report

Comments and Suggestions for Authors

The authors heeded the suggestions

Reviewer 3 Report

Comments and Suggestions for Authors

Perfect! Based on the author's response to my report, the authors have carefully addressed all my comments in the revised manuscript. In my opinion, there are no further comments.